# Bird tolerance to humans in open tropical ecosystems

Peter Mikula [1,2,3,4] ✉, Oldřich Tomášek [1], Dušan Romportl[5], Timothy K. Aikins [6,7], Jorge E. Avendaño[8,9], Bukola D. A. Braimoh-Azaki [10,11], Adams Chaskda[11], Will Cresswell [12], Susan J. Cunningham [7], Svein Dale[13], Gabriela R. Favoretto [14], Kelvin S. Floyd[15], Hayley Glover[16], Tomáš Grim [17], Dominic A. W. Henry[18], Tomas Holmern[19], Martin Hromada [20,21], Soladoye B. Iwajomo [22,23], Amanda Lilleyman[24], Flora J. Magige[25], Rowan O. Martin [7,26], Marina F. de A. Maximiano[27], Eric D. Nana[28], Emmanuel Ncube [29], Henry Ndaimani[30], Emma Nelson[31], Johann H. van Niekerk[32], Carina Pienaar [33], Augusto J. Piratelli [34], Penny Pistorius[7], Anna Radkovic[16], Chevonne Reynolds [7,35], Eivin Røskaft [19], Griffin K. Shanungu [15,36], Paulo R. Siqueira[37], Tawanda Tarakini [29,38], Nattaly Tejeiro-Mahecha[39,40], Michelle L. Thompson[7], Wanyoike Wamiti [41], Mark Wilson[42], Donovan R. C. Tye[43], Nicholas D. Tye[44], Aki Vehtari [45], Piotr Tryjanowski [46,47,48], Michael A. Weston[16], Daniel T. Blumstein [4] & Tomáš Albrecht [1,2]

Animal tolerance towards humans can be a key factor facilitating wildlife–human coexistence, yet traits predicting its direction and magnitude across tropical animals are poorly known. Using 10,249 observations for 842 bird species inhabiting open tropical ecosystems in Africa, South America, and Australia, we find that avian tolerance towards humans was lower (i.e., escape distance was longer) in rural rather than urban populations and in populations exposed to lower human disturbance (measured as human footprint index). In addition, larger species and species with larger clutches and enhanced flight ability are less tolerant to human approaches and escape distances increase when birds were approached during the wet season compared to the dry season and from longer starting distances. Identification of key factors affecting animal tolerance towards humans across large spatial and taxonomic scales may help us to better understand and predict the patterns of species distributions in the Anthropocene.

Open tropical ecosystems such as savannahs, grasslands, and shrublands are globally extensive, encompassing many emblematic and iconic life forms[1]. The biota in these ecosystems form an essential component of global biodiversity. Despite being crucial for human livelihoods, these ecosystems are increasingly threatened by increasing human demands for resources. Their exploitation leads to habitat degradation, fragmentation, pollution, land conversion of natural areas through agriculture, pastoralism, hunting, extensive tourism, and other anthropogenic influences, including climate change[2,3]. Human-induced environmental changes are significant threats to biodiversity on Earth, driving widespread and substantial population declines and local extinctions of animals in the wild[4]. Human activities

have modified many habitats and ecological communities and will continue to do so throughout the Anthropocene. Consequently, animals, including those in open tropical ecosystems, will increasingly occur and interact under novel abiotic and biotic conditions that differ from those under which they evolved. Hence, there is an urgent need for research that can be directly translated to wildlife management and conservation practices.

Behaviour is an important mechanism by which animals flexibly cope with environmental challenges, including environmental variation[5]. Prey animals have evolved multiple defensive strategies and escape is one of the most important mechanisms by which they can reduce the probability of becoming prey[6]. Although timid behaviour may act as a buffer against predators, it may be maladaptive in other contexts. Animals often perceive humans as a threat even when their mutual interactions are non-lethal, and anthropogenic stimuli may trigger behavioural and physiological reactions analogous to those evoked by real predators[7]. With increasing human population pressure, particularly in open tropical areas, animals with increased fearfulness and responsiveness to humans and anthropogenic stimuli may pay high costs from human-induced disturbance through increased metabolic costs and production of stress hormones, and deteriorated immune function, foraging efficiency, reproductive success and survival with possible cascading effects on population sizes[8,9]. For instance, European and Australian birds with declining populations are less tolerant to an approaching human than birds with increasing populations[10], indicating that a level of tolerance of animals towards humans may be one of the crucial mechanisms in wildlife–human coexistence[11].

Previous studies on this topic, however, have focused mostly on parts of Europe, North America and Australia[11,12], leaving the tropics largely understudied. Tropical and temperate regions markedly differ in many aspects; for example, predation risk by natural predators is higher in tropical regions[13] and also extensive hunting pressure by humans may be higher in the tropics[14]. Predation strongly affects bird life histories either directly or indirectly. This may cause tropical birds to have typically smaller clutch sizes[15], be more risk-averse[16], and also live longer[17] than their temperate zone counterparts. Identifying the traits and behavioural mechanisms that would help us predict how tropical species will respond to anthropogenic stimuli could therefore have important benefits for wildlife conservation.

Here, we present a comprehensive assessment of the tolerance of birds towards humans in 953 species (120 families and 32 orders), representing more than one third of all bird species occurring in the open tropical ecosystems of three continents, Africa, South America and Australia (Fig. 1). We particularly aimed to identify key life-history traits and environmental variables that best predict the direction and magnitude of tolerance of open tropical ecosystem birds towards humans. We estimated the level of tolerance towards humans with a simple method, measuring their flight initiation distance, i.e., the distance at which birds escape when approached by a human observer under standardised conditions[12,18,19]. Longer flight initiation distance can be interpreted as signs of less tolerant (or shy and risk-averse) behaviour whereas shorter escape distances indicate more tolerant (or bold and risk-taking) behaviour. Ideally, flight initiation distances are used for setting buffer zones to mitigate adverse effects of human visitors on wildlife[20,21] and should be of wide interest for conservation managers, policymakers, land-use planners, and wildlife and community ecologists. To our knowledge, this study represents the first attempt to comprehensively describe and explore spatial and cross-species circumtropical variation in wildlife tolerance towards humans and it provides results that could contribute to evidence-based conservation management.

Our results suggest that some patterns in birds' tolerance towards human disturbance may be universal, such as earlier escapes in larger birds or when approached from longer initial distances and in areas with lower human disturbance, whereas other associations may show higher geographic, taxonomic or temporal variation.

## Results

We first selected two proxies for the level of human disturbance at each sampled site, specifically habitat type (rural or urban) and the level of human disturbance (measured as human footprint index which represents cumulative human pressure based on variables such as built-up environments, human population density, or infrastructure density)[12,22]. Then, we collected data on a set of ecological and environmental variables and life-history traits that correlated with avian tolerance towards humans (i.e., flight initiation distance) in previous studies. These included starting distance (i.e., the distance from which a human intruder started to approach a focal bird), body mass, clutch size, wing shape, migratory behaviour, flock size, season, ground foraging, tree cover, continent, altitude, and latitude (for details, see Supplementary Methods). We then employed Bayesian phylogenetically- and spatially informed regression analyses to test for association between avian tolerance and these predictors, either using the full set of species or a subset of passerine birds (Order: Passeriformes), which form the largest but still relatively uniform radiation of extant birds and are often studied in ecological and conservation research.

Our analysis covering the full set of species revealed that the level of avian tolerance towards humans was associated with habitat type, human footprint index, starting distance, body mass, clutch size, wing shape, and season (Fig. 2 and Supplementary Table 1). Avian tolerance was lower (i.e., escape distances were longer) in birds inhabiting rural habitats (when compared with urban habitats) and in areas with a lower human footprint index. Because habitat type and human footprint were relatively strongly intercorrelated, we also fitted models where only one of these variables was included; these models again revealed that both habitat type and human footprint were associated with avian tolerance (Supplementary Table 1). Moreover, lower tolerance was also detected in birds when approached from longer starting distances, with larger body mass, clutch size and elongated wings. Birds were also more risk-averse when approached during the wet season compared to the dry season.

We then re-ran these analyses for the subset of passerine birds and, similarly to the full dataset, found that passerines were less human-tolerant in rural areas, during the wet season, with increasing body mass, and when approached from longer starting distances (Fig. 2 and Supplementary Table 1). Remarkably, we detected no association between human tolerance by passerine birds and the human footprint index. However, models where only either habitat type or human footprint were included revealed that avian tolerance also increased with increasing human footprint (Supplementary Table 1). Finally, we found that tolerance was lower in long-distance migratory passerines, which were mainly temperate zone migrants that overwinter in open tropical ecosystems–compared to sedentary species, which were mostly tropical species.

Finally, we re-fitted the model using a subset of all species that were sampled in both rural and urban habitats. We found that rural and urban populations of the same species still significantly differed in their escape responses, with rural birds generally showing lower tolerance towards humans (Supplementary Table 1).

## Discussion

Our data of hundreds of bird species and populations inhabiting open tropical ecosystems of three continents (Africa, South America and Australia) showed that birds' tolerance towards human disturbance (measured as flight initiation distance) was best predicted by the human intruder's starting distance[19,23], the degree of exposure to human activity in terms of urbanisation and a human footprint index[11,12], but also some life-history traits such as body mass[18] and the season[23]. Avian tolerance towards humans decreased (i.e., escape

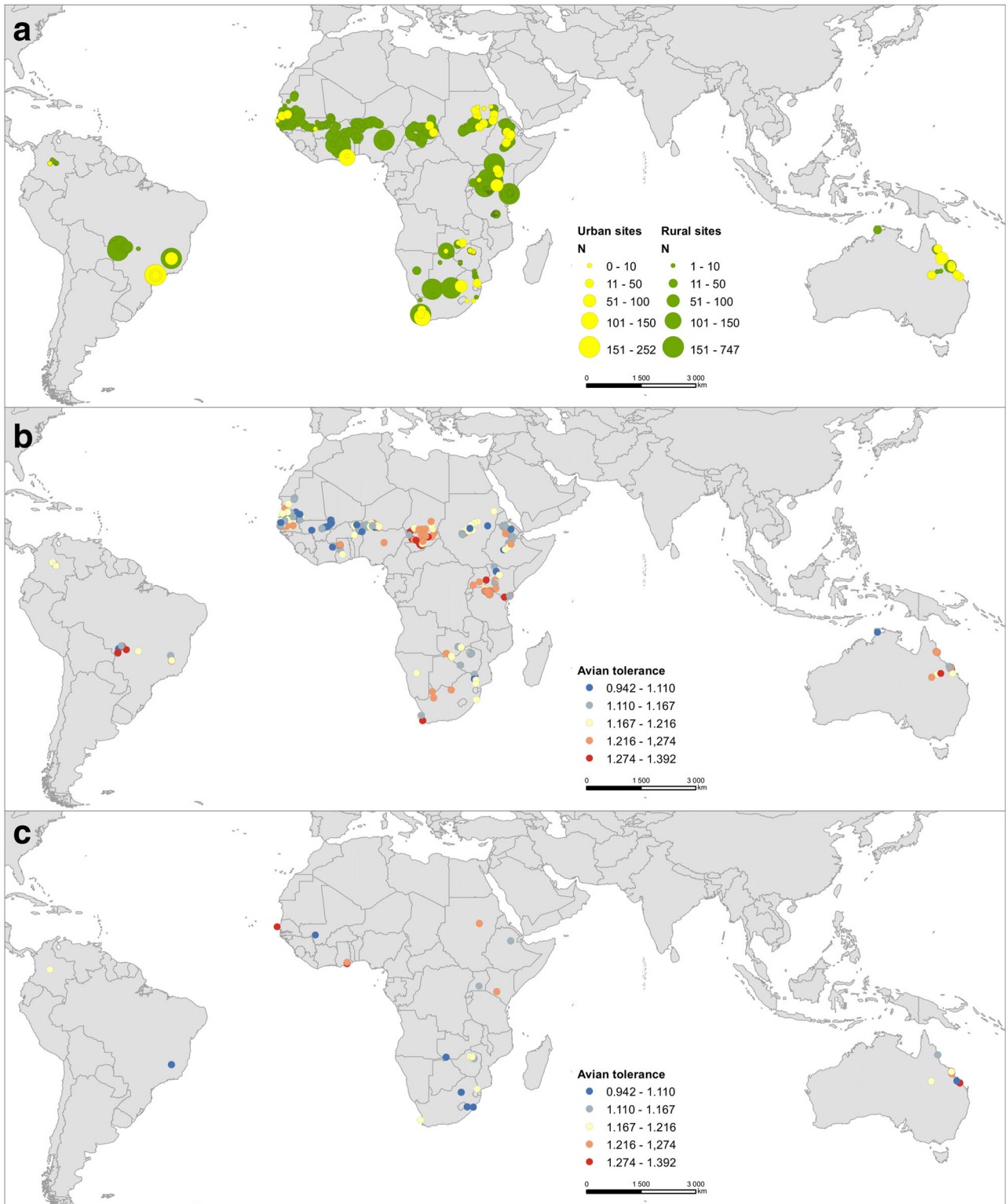

**Fig. 1 | Sampling effort and avian tolerance towards humans. a** Number of observations at rural (green colour) and urban (yellow colour) sites; the sample size is indicated by circle size. **b** Avian tolerance towards humans across rural and **c** urban sites. Tolerance towards humans by birds was estimated as residual variance in the flight initiation distance per each site from the main model. Red shades indicate lower tolerance of birds towards approaching humans (i.e., birds had longer escape distances), whereas blue shades indicate the opposite. Note that some very nearby urban and rural sites shared the same geographic coordinates−for clarity, these sites were excluded from (**b, c**), respectively. The maps were created using open data on country boundaries of the world (source: public.opendatasoft.com, Open Government License v3.0) and data acquired and processed by the authors of the paper in ArcGIS Pro software (Environmental Systems Research Institute, Inc., Redlands, CA, 2022).

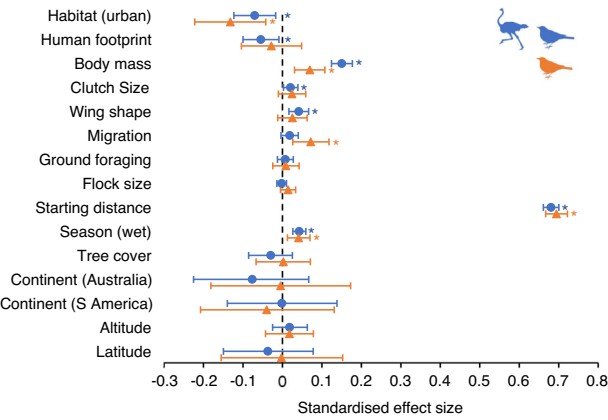

**Fig. 2 | Results of multivariate Bayesian phylogenetically- and spatially informed regressions.** We evaluated the association between avian tolerance towards humans (measured as the flight initiation distance; dependent variable) and several life-history and environmental predictors across birds of open tropical ecosystems (all species: blue colour, $N = 10{,}249$ observations for 842 species; passerines: orange colour, 5400 observations for 425 species) and reported standardised effect sizes (coloured objects) with their 95% credible intervals (horizontal lines). Predictors included habitat type (rural or urban), human footprint index, body mass, clutch size, wing shape (measured as hand-wing index), presence of migratory behaviour, ground foraging, flock size, starting distance, season (wet or dry), percentage tree cover, continent (Africa, Australia or South America), altitude and latitude. We considered an association significant if the credible intervals did not overlap zero—statistically significant results are highlighted by "*". For information on sample sizes and full statistical results, see Supplementary Table 1. Bird silhouettes were downloaded from PhyloPic (http://phylopic.org) and are available under the Public Domain Dedication 1.0 license (https://creativecommons.org/publicdomain/zero/1.0/).

distance increased) in areas with lower human disturbance and during the wet season of the year. In addition, birds flushed at longer distances with increasing starting distances from which humans initiated their experimental approach, and tolerance was lower in larger birds. Finally, we found that some other factors were significantly associated with avian tolerance either when considering all species (wing shape and clutch size) or the passerine clade (migratory behaviour) only.

Our results revealed that birds exhibited increased tolerance in urban habitats and in sites with high human footprint index (in passerines, the effect size was similar to the analysis of all species but the credible interval crossed zero in the full model). Moreover, rural–urban differences in avian tolerance persisted even when we restricted our analyses only to species sampled in both habitat types. These results indicate that pronounced human presence and disturbance modifies birds' risk assessment. Indeed, this pattern is routinely described in the literature, but most previous large-scale comparisons are based on birds from Western and Central Europe[12,24]. However, our results show that this pattern can also be generalised to birds in tropical regions. Three main non-mutually exclusive intraspecific mechanisms may explain why human-tolerant individuals are common in areas with significant human activity: non-random personality-dependent habitat preferences whereby bolder individuals are more likely to enter human-disturbed habitats; within generational plasticity (i.e., habituation-like process drive behavioural flexibility); and evolutionary selection and local adaptation.

First, available studies on marked individuals have shown high individual consistency and cross-generation heritability of avian tolerance towards humans[25,26]. As a consequence, individual birds may not be able to adjust their tolerance plastically to the level of human disturbance. Instead, variation in behavioural syndromes and inherent levels of susceptibility to human disturbance may affect preferences to settle in areas with different levels of human disturbance, whereby

bold individuals come to occupy more human-disturbed areas and shy individuals preferentially settle in areas with lower disturbance[25].

Second, the widespread and presumably multiple independent and rapid origins of avian tolerance across the human-disturbed open tropical ecosystems, spanning different taxonomic groups and geographic regions with different urbanisation history and patterns, may indicate that behavioural plasticity is potentially important mechanism behind the increased tolerance of birds towards humans[11,23]. Increasing human presence in global ecosystems may, directly and indirectly, trigger flexible behavioural changes in animal behaviour. Direct effects may include a situation when high human presence in urbanised habitats drives urban animals to escape at shorter distances that approach a boundary. Indirect effects may involve lower activities of predators in urban areas, decreasing animal fearfulness[12].

Finally, increased tolerance in disturbed habitats may also represent a local adaptation[27] at the population level. It is possible that the evolution of traits promoting higher tolerance towards humans may emerge in animal populations that coexist in close proximity to humans over long time periods. Altogether, the results of the analysis restricted to the same species and populations occurring in both rural and urban habitats are consistent with mechanisms acting at the intraspecific level. Unfortunately, we are not able to distinguish between the particular intraspecific mechanisms because, like many other similar studies, sampled birds were not marked.

Our results also cannot rule out species-level mechanisms. It is essential to recognise that birds which can tolerate increased levels of disturbance and invade human-dominated areas are typically a non-random sample of species available in a regional species pool, and many birds are unable to inhabit human-dominated areas[28,29]. Avian presence in human-dominated areas may also have mechanistic explanations, such as species commonness in natural habitats[30]. However, the appropriate adaptations for anthropogenically altered environments, including enhanced behavioural plasticity and cognitive skills, seem to be crucial for the successful coexistence of wildlife with humans[31–33]. Altogether, to improve species conservation and management, it is important that future studies will distinguish between species-level and intraspecific mechanisms facilitating wildlife–human coexistence.

Interestingly, both binary habitat type and a continuous human footprint index were significant predictors of avian tolerance towards humans, indicating that the two variables capture slightly different aspects of human disturbance. Future studies focusing on different aspects of human disturbance at sites and animal tolerance towards humans may shed a new light on the primary drivers of increased tolerance of wildlife towards humans in disturbed habitats.

In this study, the strongest predictor of avian tolerance was starting distance of the approaching human[19,23,34,35]. This demonstrates that birds found in open tropical ecosystems assess risk dynamically, escaping significantly earlier when approached from longer starting distances[34]. This is in agreement with the Flush Early and Avoid the Rush (FEAR) hypothesis[36] which predicts that animals should initiate their escape early after spotting and beginning to monitor an approaching threat to avoid excessive attentional costs of ongoing monitoring (e.g., in terms of physiological costs and decreased foraging activity)[37]. Alternatively, birds may assess a greater risk if approached for a longer time (i.e., from a longer distance). This means that to properly quantify the mean and the variance in escape distances of birds and set up appropriate buffer zones[20,21], it is necessary to approach individuals from various distances to estimate whether and, if so, the distance at which the relationship between the flight initiation and starting distance plateaus[34,38]. Identifying this may be particularly useful if seeking to identify a maximum escape distance to define buffer zones and adopt appropriate conservation practices such as providing hides from which to observe birds in relatively small sites.

We found that two biometric traits, body mass and wing shape, were correlated with avian tolerance towards humans. There is robust empirical evidence that body mass is one of the best predictors of tolerance and risk-taking across animal species with larger animals typically escaping earlier[12,18,19,39]. Body mass is a crucial life-history trait determining pace-of-life syndromes in animals through its tight correlation with lifespan, adult mortality, reproduction patterns, and metabolic rates[17,40]. Larger animals typically live longer and prioritise high survival and future over current reproduction (slow pace-of-life)[41]. However, larger birds have elevated extinction risks[4] and their exposure to harmless human activities may make them more vulnerable to legal and illegal hunting and to predation by natural predators. If so, this creates a potential ecological trap for larger and more tolerant birds. Hence, the costs and benefits of increased tolerance of birds towards humans must be considered together when designing and managing ecotourism activities in focal areas.

We also found that birds with elongated wings (in the full dataset) and long-distance migratory birds (in the passerine subset) escaped earlier than shorter-winged and non-migratory/resident species, respectively[42,43]. Elongated wings reduce the costs of flight[44] and increased selection on flight efficiency is also present in long-distance migratory species[45]. Efficient fliers may thus escape earlier because the relative costs of escape are lower for them than for species with less efficient flight. In addition, migratory (typically temperate) species may be less familiar with local environments at their tropical wintering grounds than sedentary species which may increase risk-aversion in migratory species[46].

Our finding that birds were less tolerant of human approach during the wet than the dry season may indicate that tolerance is temporally variable in open tropical ecosystem birds, presumably reflecting changes in the relative costs and benefits of escape behaviour during the annual life cycle[23]. Breeding season is poorly described for many tropical birds but it can generally cover the wet or dry season or both; tropical birds also often breed over a prolonged season that may include much of the year[47,48]. Decreased tolerance was documented for breeding birds (when compared with other periods of year)[23], hence, lower tolerance during the wet season may indicate that many birds in our sample were breeding during that period. Risk-aversion during the breeding season may decrease the probability of death of adults from predation when providing care for their clutch or nestlings and this caution may also help avoid revealing a nest's position. Early escape has been found to be correlated with higher baseline concentrations of the stress hormone, corticosterone[49]. In birds, levels of corticosterone increase during the breeding season and this affects their dispersal propensity and potentially also their tolerance towards humans[50,51]. Alternative explanations for the effect of the season may include seasonal variation in age structure of the population (with more juveniles than adults in the dry than wet season), time of moulting, differences in the predation pressure by natural predators and hunting activity of humans, thermal constraints or different values of patches that are associated with changing resource availability.

After controlling for many confounding factors including body size, we found a weak effect of clutch size on avian tolerance in the full dataset (there was a similar trend in the passerine dataset but the credible intervals slightly crossed the zero), showing that birds with relatively larger clutches were less tolerant than species with lower investments in a single brood. This result is somewhat surprising because species with adults investing more into the current reproduction are usually predicted to be more tolerant to the human approach because their future reproductive value is lower compared to low-fecund species[13,19,52]. However, some of these predictions were evaluated by studying birds sitting on or occurring in the close vicinity of their nests[13] whereas all of our data on avian tolerance were collected further from their nests. Moreover, nest visitation and food delivery rate increase with clutch size[53]. Hence, one could again expect increased tolerance in species with larger clutches since they are expected to be under stronger pressure for foraging time and efficiency. Clearly, the mechanism(s) behind the association between off-nest escape decisions of birds and their investment in reproduction are unclear and this topic requires further study.

We found negligible associations between avian tolerance and ground foraging, flock size, tree cover, altitude, latitude, and continent in either the full dataset or the subset restricted to passerines. The lack of an association between avian tolerance and, for example, flock size might be caused by the fact that most observations in our dataset included single birds (flock size: median = 1, mean ± SD = 4.3 ± 48 individuals). Nonetheless, this insufficient sampling may not explain the absence of latitudinal variation in our data given that our dataset covered a latitudinal range that was similar to a large-scale comparative study from Europe which reported a clear increase in avian tolerance towards humans with latitude[12]. Instead, we found low latitudinal variation in this trait, perhaps because our latitudinal comparison (ranging ~30 absolute latitude degrees) was located mainly in the tropics of the Southern Hemisphere. Birds exhibit relatively low latitudinal variation in several life-history traits, including longevity[17], clutch size[15], and wing shape[45], and another trait that appears to show the same pattern is avian tolerance towards humans.

Altogether, our study emphasizes that a relatively small number of variables, which can be easily obtained in the field or literature, are powerful predictors of the magnitude and the direction of tolerance towards humans across birds in open tropical ecosystems, and this has important implications for wildlife conservation and management. For instance, such data can be used to help develop set-back zones[20,21] to protect vulnerable species and highlight that annual variation (e.g., during the wet season) might require variable management strategies throughout the year. Our results also support the idea that some patterns of avian tolerance may be general, such as earlier escapes elicited in areas with lower human disturbance and activity[11,12], earlier escapes initiated by longer approaches[19,23,35] as well as larger (and often threatened) birds being more risk-averse[10,18,22], whereas other associations may be more geographically, taxonomically or temporally variable[10,23]. To our knowledge, our study represents the first attempt to describe circumtropical variation in avian tolerance towards humans. Global conservation efforts and modelling of animal tolerance towards humans may benefit from future studies focusing on tropical communities of birds and other animals outside open ecosystems, as well as developing a deeper understanding of the mechanisms promoting tolerance towards human disturbance across animals.

## Methods

### Study sites and data sampling

We focused on open to semi-open terrestrial ecosystems which dominate many tropical areas, particularly in Africa, South America and Australia (Fig. 1), and are characterised by high climatic seasonality. Our field data included different open tropical ecosystem subtypes, from semideserts, grasslands, shrublands, and arid savannahs of Sahel region and Southern Africa to moist savannahs and wetlands in South America. In Africa, the fieldwork was conducted between 2002 and 2021; African sampling spanned much of the continent, from Senegal and Mauritania in the West, to Kenya and South Africa in the East and South, respectively. In South America, data were collected between 2011 and 2021 in the Cerrado and Pantanal ecoregions (Brazil), and the Llanos ecoregion (Colombia). In Australia, data were collected in 2000 and again between 2011 and 2017 in Queensland and in the Northern Territory. All field measurements were time- and georeferenced in the field (using GPS) or later using coordinates from Google Maps.

Field data came from many observers and world regions and were typically collected before the start of this project. Hence, sites in this study differ in their size but, we understand that sites are continuous

areas with relatively a homogeneous habitat. We recognize this assumption is somewhat subjective given the number of people who collected these data, mostly for other studies. Some observers assigned unique geographic coordinates to each observation; in these cases, we clumped nearby observations from the same-type environment under a single site.

## Avian tolerance towards humans

Flight initiation distance constitutes a reliable measure of an animal's willingness to take a risk and their tolerance towards human disturbance, reflecting the trade-off between the fitness-related benefits of staying and the costs of escaping[18,39]. All escape data were collected using standard procedure[19,34]. Briefly, when a focal bird was spotted, a single observer moved at a normal walking speed (-1 m s[-1]) directly towards the bird (with head oriented towards the bird and maintaining eye contact). The flight initiation distance was estimated as the distance (estimated by a number of -1 m steps, conversion of a number of steps to metres, or using a rangefinder) between the position of the approaching observer and focal bird when the bird initiated the escape. When a focal bird was positioned on a perch (e.g., vegetation or a human made object), the flight initiation distance was corrected for perch height, and straight distance was estimated either directly by rangefinder or calculated as the Euclidean distance (which equals the square root of the sum of the squared horizontal escape distance and the squared height of the perch). All researchers using the step method were well-trained before data collection to make their steps constantly -1 m long or to be able to convert the distance measured by steps to metres, making these data directly comparable to data collected by rangefinders. We approached only birds that showed no considerable signs of distress; relaxed birds were foraging, roosting, or preening. Observers did not approach birds at their nests. When in a flock, the flight initiation distance from one randomly chosen individual was measured although the reaction of the selected bird individual might be affected by behaviour of other birds in the flock. Researchers wore outdoor clothing with no bright colours during data collection. We attempted to minimise resampling individuals by not sampling the same site repeatedly although even a modest degree of resampling individuals should not be problematic[54]. The majority of data were gathered in the morning (06:00–10:00) and afternoon (15:00–18:00) when birds were most active (~76% of all observations). All researchers were trained to measure escape distances using standardised protocols; previous research found that these estimates used to be highly repeatable among observers[55]. Flight initiation distance estimates are also highly consistent for individuals, populations and species under similar contexts[12,20,23]. All flight initiation distance data were collected blindly with respect to the tested hypotheses, hence preventing any conscious or unconscious bias. Altogether, we collected 14,998 flight initiation distances for 953 bird species (120 families and 32 orders) (Fig. 1). However, this sample was reduced to 10,249 observations for 842 species (in full dataset; for details, see below) and 5,400 observations for 425 species (in the dataset for passerines), respectively, because some predictor values were missing for some species.

## Predictors

Observers recorded data for each bird that included starting distance (in metres; the distance between the initial position of the human observer and the position of bird when first spotted and started to be approached by an observer) and flock size (the number of all bird individuals moving, feeding, or perching together; observers typically approached only single-species flocks). Species-specific body masses were extracted from EltonTraits 1.0 database[56]. Data on clutch sizes were retrieved from ref. 57. As an index for wing shape and a general estimate of flight ability and efficiency, we used the hand-wing index from ref. 45; species with higher hand-wing index

have narrower and elongated wings suitable for long-distance flight, whereas species with lower hand-wing index have broader wings suitable for short-distance flight or are associated with weaker flight performance. Migratory behaviour was coded as 0 for tropical sedentary, nomadic, and altitudinal migrant species, and 1 for long-distance (temperate) migrants, using data from BirdLife's database (for the definition of each category, see Supplementary Methods)[58]. We defined the wet season as months when the mean monthly average was greater than the year-round mean (otherwise, an observation was assigned to the dry season) using data from the Climate Change Knowledge Portal (https://climateknowledgeportal.worldbank.org). We calculated a ground foraging index which equals the proportion of foraging time spent on the ground or water when compared with the time spent elsewhere (understory, mid-story, canopy, and air) using data in EltonTraits[56].

Several environmental and geographic variables that may influence avian tolerance towards humans were estimated for all sampled locations. These included tree cover (available at https://data.globalforestwatch.org/)[59], altitude (available at https://earthexplorer.usgs.gov/; United States Geological Survey), as well as indicators of anthropogenic disturbance captured by the 2009 human footprint index (available at https://sedac.ciesin.columbia.edu/data/set/wildareas-v3-2009-human-footprint)[60,61]. Each site was assigned to single geographic coordinates; site-specific values of these three variables were calculated as the mean value for a 2-km radius buffer zone. All geographic, spatial and habitat analyses were processed in environment ArcGIS and associated extensions and toolboxes (e.g., Spatial Analyst). Each site was also assigned to the habitat type (0 = rural: areas with natural or agricultural landscape with no or sparsely located buildings; 1 = urban: areas with continuous urban elements like multi-storey buildings, family houses or roads) directly during the fieldwork, and continent. Habitat type is a more subjective proxy of human disturbance at sites than the human footprint index. However, habitat type is a long-established and widely-used proxy for the level of human disturbance in studies on escape behaviour of birds[12,25] and the two indexes may differ in some aspects of human disturbance variation they capture. For further details on predictor variables and justification for their use, see Supplementary Methods.

**Phylogenetic tree construction.** We combined the data on avian tolerance measured as the flight initiation distance and other bird traits with a time-calibrated phylogeny generated from the online tool available at http://birdtree.org/ [62]. We downloaded 1000 trees using the Hackett backbone. We reconstructed a maximum clade credibility tree from these 1000 trees using function maxCladeCred in phangorn package (version 2.8.1)[63].

**Statistical analyses.** We analysed these data using Bayesian models with Hamiltonian Monte Carlo sampling built in the probabilistic language Stan through the CmdStanR (version 0.4.0) interface and using posterior package (version 1.1.0) in R software (version 4.1.2)[64–66]. The flexibility of Stan enabled us to control for both the phylogenetic and spatial autocorrelation in the data by modelling them as latent Gaussian processes. We modelled phylogenetic covariation among species as a Gaussian process with Ornstein-Uhlenbeck covariance function $K_{OU(i;j)} = \eta_P^2 \exp(-D_{Pij}/\rho_P) + \delta_{ij}\eta_P^2$, where $D_{Pij}$ is the phylogenetic distance between species $i$ and $j$, $\eta_P$ is the marginal deviation determining the maximum covariance between species, $\rho_P$ is the length-scale parameter controlling how quickly the correlations fade with time (i.e., with the phylogenetic distance), and $\delta$ is the Kronecker delta. To model spatial covariance, we used a squared exponential covariance function $K_{SE(k;l)} = \eta_S^2 \exp(-D_{Skl}/\rho_S^2) + \delta_{kl}\eta_S^2$, where $D_{Skl}$ is spatial distance between sites $k$ and $l$, $\eta_S$ is the marginal deviation determining the maximum covariance between sites, and $\rho_S$ is the length-scale parameter controlling how quickly the correlations fade with spatial

distance[67]. The diagonal of both the phylogenetic and the spatial covariance matrix also included an additional variance term for an unstructured variance among species and sites, respectively. All models also included varying intercepts of species, years, sites, and data collectors.

Migration tendency, habitat, season, and continent were fitted as categorical predictors, whereas all the other variables as continuous predictors. Flight initiation distance, starting distance, body mass, clutch size, and flock size were log-transformed and all continuous variables were standardised by dividing them by two standard deviations before fitting the models to obtain standardised effect sizes in the form of standardised partial regression coefficients[68]. Dividing by two standard deviations ensures the comparability of the effects of continuous and categorical predictors[69]. Prior to the regression analyses, we checked the correlation between predictors, revealing generally low multicollinearity with the exception of the correlation between habitat type and human footprint index ($r = 0.55$ and 0.62 for the full and the passerine dataset, respectively; Supplementary Figs. 1 and 2). First, we fitted a full model including all predictors. Full model was fitted separately for all birds and for the passerine clade. Second, we fitted full models (again for all species and passerines only) excluding either habitat type or human footprint index from the set of predictors. Third, we fitted full models for all species and passerines only using a subset of all species occurring in both rural and urban habitats; if same-species individuals had similar escape responses in the two habitats, this would suggest fixed evolutionary constraints at the species level.

To prevent overfitting due to a relatively high number of predictors, we used a scaled prior, which was defined for all predictor parameters as having zero mean and standard deviation $\sqrt{R_p^2/k}\sigma_y$, and the residual error as having zero mean and standard deviation $\sqrt{1 - R_p^2}\sigma_y$, where $R_p^2$ is a prior belief about $R^2$, $k$ is the number of predictors, and $\sigma_y$ is a standard deviation of the response (the latter equals to one given the use of standardised variables). To test the sensitivity of the models to the prior, we fitted models with scaled priors assuming $R_p^2$ to be 0.1, 0.2, 0.3, 0.5, 0.7, and 0.9, respectively. The results were robust regardless of the prior used. In the main text, we only report results with scaled priors for $R_p^2 = 0.3$ (Supplementary Table 1), which seems to be a reasonable prior belief about the proportion of variance explained by the models, given that the models included several predictors previously shown to be associated with escape distance of birds. The priors for length-scale parameters of both Gaussian processes were set as the inverse-gamma distribution with shape parameter $\alpha = 1.5$ and scale parameter $\beta = 0.057$. The resulting prior distribution minimises the probability of values lower or higher than the observed standardised phylogenetic or spatial distances (with the maximum values equal to one). The models were sampled in twelve chains, each with 1000 warm-up and 3000 sampling iterations, and thinning set to 5. Potential scale reduction factor was <1.01 in all cases, indicating good convergence of the inference[70].

### Reporting summary
Further information on research design is available in the Nature Portfolio Reporting Summary linked to this article.

## Data availability
All data used in this study are available at the Open Science Framework repository (https://doi.org/10.17605/OSF.IO/BSPQX).

## Code availability
All codes used to generate the results in this study is available at the Open Science Framework repository (https://doi.org/10.17605/OSF.IO/BSPQX).

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

## Acknowledgements

We are especially thankful to Rob G. Bijlsma who generously shared with us his extensive dataset from the Sahel region. We are also thankful to Afan Ajang, Linn M. Bjørvik, Tamuka Chapata, Wouter van Dongen, Patrick Guay, Lenka Harmáčková, Lukasz Jankowiak, Jan van der Kamp, Lennox Kirao, Jakub Kosicki, Philista Malaki, Pretty Maoko, John Mchetto, Grayson Mwakalebe, Organisation for Tropical Studies (South Africa), Diogo Samia, Trine Hay Setsaas, Libor Vaicenbacher and Leo Zwarts for their help with data collection. MW is thankful to Allison Piper, and a Deakin University Faculty of Science, Engineering and the Built

Environment National and International Research Collaboration Grant in Kenya and BEACH (Beach Ecology and Conservation Hub; Venus Bay) in Australia. In Kenya, field data collection was approved by National Commission for Science Technology and Innovation no. NACOSTI/P14/4653/660 to M.W. and P.M. and no. NACOSTI/P18/52438/25493 to MW, Kenya Wildlife Service no KWS/BRP/5001 to M.W. A Rocha Kenya and the National Museum of Kenya supported and helped conduct fieldwork in Kenya. In South Africa, the University of Cape Town Science Faculty Animal Ethics Committee (2015/V11/SC) to S.J.C., Northern Cape Department of Environment and Nature Conservation (FAUNA 1489/2015) to P.P. In Brazil, we worked on private lands where no permits were required. In Australia, research was approved by the Deakin University (B32/2012, B11/2015, B10/2018), the Charles Darwin University Animal Ethics (A11013), the Macquarie University Animal Research Committee (99021), the Queensland Parks and Wildlife Service (#FA/000379/00/SA), and the Northern Territory Parks and Wildlife (41035 and 55233). This study was financially supported by the DSI-NRF Centre of Excellence at the FitzPatrick Institute of African Ornithology, University of Cape Town (grant to S.J.C.), The Leventis Foundation through the A.P. Leventis Ornithological Research Institute, Jos Nigeria (grant to B.D.A.B.), by a fellowship of the Fulbright (Slovakia) programme to P.M. for a visit to the University of California, Los Angeles.

## Author contributions

P.M., O.T., D.R., P.T., M.A.W., D.T.B. and T.A. formed the core team behind this work. P.M. conceived and designed the project. P.M., T.K.A., J.E.A., B.D.A.B., A.C., W.C., S.C., S.D., G.R.F., K.S.F., H.G., T.G., D.A.W.H., T.H., M.H., S.I., A.L., F.J.M., R.O.M., M.F.A.M., E.D.N., E.Nc., H.N., E.Ne., J.H.N., C.P., A.J.P., P.P., A.R., C.R., E.R., G.K.S., P.R.S., T.T., M.L.T., W.W., M.W., N.T.M., D.R.C.T,. N.D.T., P.T., M.A.W., D.T.B. and T.A. collected field data. P.M. compiled the life-history data, D.R. extracted environmental data. O.T., P.M. and T.A. designed the analyses with the help of D.T.B. and M.W. O.T. wrote scripts and ran the analyses with input from A.V. D.R. prepared map visualisations. P.M. wrote the first version of the manuscript with inputs from D.T.B., M.A.W., T.A., O.T. and P.T. All authors edited and approved the manuscript.

## Competing interests

The authors declare no competing interests.

## Additional information

¹Institute of Vertebrate Biology, Czech Academy of Sciences, Květná 8, 603 65 Brno, Czech Republic. ²Department of Zoology, Faculty of Science, Charles University, Viničná 7, 128 44 Praha 2, Czech Republic. ³Faculty of Environmental Sciences, Czech University of Life Sciences Prague, Kamýcká 129, 165 00 Prague, Czech Republic. ⁴Department of Ecology and Evolutionary Biology, University of California, 621 Young Drive South, Los Angeles, CA 90095-1606, USA. ⁵Department of Physical Geography and Geoecology, Faculty of Science, Charles University, Albertov 6, 128 43 Prague 2, Czech Republic. ⁶Department of Biodiversity Conservation and Management, University for Development Studies, P.O. Box TL 1882 Tamale, Ghana. ⁷FitzPatrick Institute of African Ornithology, DSI-NRF Centre of Excellence, University of Cape Town, Rondebosch 7701, South Africa. ⁸Laboratorio de Biología Evolutiva de Vertebrados, Departamento de Ciencias Biológicas, Universidad de los Andes, Bogotá, Colombia. ⁹Programa de Biología, Universidad Distrital Francisco José de Caldas, Bogotá, Colombia. ¹⁰Department of Biological Sciences, University of Cape Town, Cape Town, South Africa. ¹¹AP Leventis Ornithological Research Institute, University of Jos, Jos, Nigeria. ¹²Centre for Biological Diversity, University of St Andrews, St Andrews, Fife KY16 9TH, UK. ¹³Department of Ecology and Natural Resource Management, Norwegian University of Life Sciences, P.O. Box 5003 Norwegian 1432 Ås, Norway. ¹⁴Grupo de Pesquisa e Conservação da Arara-azul-de-lear, Bahia, Brazil. ¹⁵International Crane Foundation/Endangered Wildlife Trust (ICF/EWT Partnership), P. O Box 33944 Lusaka, Zambia. ¹⁶School of Life and Environmental Sciences, Faculty of Science, Engineering and the Built Environment, Deakin University, 221 Burwood Hwy, Burwood, VIC 3125, Australia. ¹⁷Department of Biology and Ecology, University of Ostrava, Chittussiho 10, 710 00 Ostrava, Czech Republic. ¹⁸Centre for Statistics in Ecology, Environment and Conservation, Department of Statistical Sciences, University of Cape Town, Rondebosch 7700, South Africa. ¹⁹Department of Biology, Norwegian University of Science and Technology, NTNU, NO-7091 Trondheim, Norway. ²⁰Laboratory and Museum of Evolutionary Ecology, Department of Ecology, Faculty of Humanities and Natural Sciences, University of Prešov, 17. novembra 1, 081 16 Prešov, Slovakia. ²¹Faculty of Biological Sciences, University of Zielona Góra, Prof. Z. Szafrana 1, 65-516 Zielona Góra, Poland. ²²Department of Zoology, Faculty of Science, University of Lagos, Akoka, Yaba, Nigeria. ²³TETFUND Centre of Excellence in Biodiversity Conservation and Ecosystem Management, University of Lagos, Lagos, Nigeria. ²⁴Research Institute for the Environment and Livelihoods, Charles Darwin University, Darwin, NT 0909, Australia. ²⁵Department of Zoology and Wildlife Conservation, University of Dar es Salaam, P.O. Box 35064 Dar es Salaam, Tanzania. ²⁶Africa Conservation Programme, World Parrot Trust, Glanmor House, Hayle TR27 4HB, UK. ²⁷Programa de Pós-Graduação em Ecologia, Instituto Nacional de Pesquisas da Amazônia. Avenida André Araújo, 69067-375 Manaus, AM, Brazil. ²⁸Institute of Agricultural Research for Development (IRAD), 1st Main road Nkolbisson – Yaoundé, Yaoundé, Cameroon. ²⁹Department of Wildlife Ecology and Conservation, Chinhoyi University of Technology, P Bag 7724, Chinhoyi, Zimbabwe. ³⁰International Fund for Animal Welfare, 22 Airdrie Road, Estlea, Harare, Zimbabwe. ³¹School of Medicine, Institute of Life Course and Medical Sciences, Faculty of Health and Life Sciences, University of Liverpool, Ashton Street, L69 3GS Liverpool, UK. ³²Department of Environmental Sciences, College of Agriculture and Environmental Sciences, University of South Africa, PO Box 392 Pretoria 0003, South

Africa. [33]BirdLife South Africa, Isdell House, 17 Hume Road, Dunkeld West 2196 Gauteng, South Africa. [34]Departamento de Ciências Ambientais, Universidade Federal de São Carlos, Rodovia João Leme dos Santos km 110, 18086-330 Sorocaba, SP, Brazil. [35]School of Animal, Plant and Environmental Sciences, University of the Witwatersrand, Private Bag 3, Wits, 2050 Johannesburg, South Africa. [36]Department of Biology, Vrije Universiteit Brussel, Pleinlaan 2, 1050 Brussels, Belgium. [37]Department of Genetics, Ecology and Evolution, Federal University of Minas Gerais, Presidente Antônio Carlos avenue 6627, 31270-901 Belo Horizonte, Brazil. [38]Research and Education for Sustainable Actions, 9934 Katanda, Chinhoyi, Zimbabwe. [39]Grupo de investigación ECOTONOS, Facultad de Ciencias Básicas e Ingeniería, Universidad de Los Llanos, Villavicencio, Colombia. [40]Colecciones Biológicas, Instituto de Investigación de Recursos Biológicos Alexander von Humboldt, Villa de Leyva, Boyacá, Colombia. [41]Zoology Department, National Museums of Kenya, Museum Hill Rd.P.O. BOX 40658- 00100 Nairobi, Kenya. [42]British Trust for Ornithology, University of Stirling, Stirling FK9 4LA, UK. [43]Organisation for Tropical Studies, PO Box 33 Skukuza 1350, South Africa. [44]C4 EcoSolutions, Tokai 7966 Cape Town, South Africa. [45]Department of Computer Science, Aalto University, PO Box 15400, 00076 Aalto, Finland. [46]Department of Zoology, Poznań University of Life Sciences, Wojska Polskiego 71c, 60-625 Poznań, Poland. [47]TUM School of Life Sciences, Ecoclimatology, Technical University of Munich, 85354 Freising, Germany. [48]Institute for Advanced Study, Technical University of Munich, 85748 Garching, Germany. ✉e-mail: petomikula158@gmail.com

