## [Peer Review File · Nature Communications]

REVIEWER COMMENTS

Reviewer #1 (Remarks to the Author):

This study tests the correlates of fear reactions of birds towards human observers (measured as flight initiation distance, FID). The study is based on a large dataset collected by the standard procedure for estimating FID. The analyses are carefully conducted and control for the potential biases due to phylogenetic relatedness between species and spatial autocorrelation. The results confirm most of the findings of earlier studies that used (geographically) different datasets, supporting that FID are predicted by some ecological and life-history variables. The study also shows that FID is a useful measure of the birds' disturbance tolerance in tropical ecosystems that have been relatively rarely covered by earlier studies. My comments below may help to improve some unclear points in the manuscript.

lines 189-191: It sounds counterintuitive to say that higher predation rates / hunting pressure in the tropics result in higher survival and longevity. Please provide a brief explanation.

l. 201-203: A longer FID indicates LESS tolerance to human disturbance while a shorter FID indicates HIGHER tolerance (i.e. the opposite of what is claimed here).

l. 221, further information on data sources in Table S1: Body mass data lacking for a species were substituted by body mass from a closely related species (Supplementary Methods l. 53-54). Then, for some species that also lack clutch size data this was substituted by data "from congeneric species with relatively similar body sizes" (Supplementary Methods l. 65-66). Does this mean that both body mass and clutch size were used from the same "donor" species? If yes, please give this information explicitly somewhere, e.g. in the Supplementary Methods.

l. 263-264: " ... based on their syndromes" -> do you mean behavioural syndromes?

l. 380-381: "All escape data were collected using standard procedure developed by ref.38." - Ref. 38 is from 2006, while data collection started in 2000 in some study sites (see l. 361-370) - thus not all data collection could follow the procedure published in 2006.

l. 396: "Researchers wore standardised outdoor clothing" - what do you mean on standardized, e.g. all observers' clothing had the same colour?

l. 437-438: "urban: areas with continuous urban elements like multi-storey buildings, family houses or roads" - What was the basis of the assessment that the sites contain these urban elements or not? Please clarify whether this was based on information from specific databases or maps.

l. 316-331, explanations for the seasonal change in FID: An important alternative explanation is that the bird communities of the studied areas change seasonally, e.g. due to the presence/absence of migrants, that may affect FID. Although both the migration status of the species and the season of the observations were included as predictors in the multi-predictor models, their effects may be difficult to separate due to their potential collinearity. Although the authors claim that multicollinearity is low among predictors (l. 470-472), this is only shown for continuous predictors (Fig S1), thus it is unclear whether season and migration status was strongly correlated (as can be assumed by the seasonal nature of migration) or not. To make the independent effect of season more convincing, I suggest to demonstrate that these predictors are not/weakly correlated. A brief discussion of the potential confounding effect of migration (especially if that is justified by its strong correlation with season) would also be useful.

Reviewer #2 (Remarks to the Author):

This study aims to assess predictors of flight initiation distance among birds in open tropical systems, with a particular emphasis on urbanization. The authors used data collection across an enormous geographic scale and species breadth. I commend the authors for their creativity and cooperation to execute such an ambitious project. Researchers often operate and publish within their respective silos; it is refreshing to see researchers from across the world combine data into a collaborative manuscript.

The manuscript is mostly well-written and clear, the research question is valid, and the methodologies appear consistent across researchers. However, I do not believe it is publishable in its current form due to the major suggestions listed below. I do believe it is possible to revise the manuscript and successfully publish it. I hope the authors take the suggestions below as constructive and use them to improve and prepare this manuscript for ultimate publication.

Major Suggestions:

1. The analyses simultaneously include correlated independent variables. This multicollinearity threatens the validity of your analyses. You say that Figure S1 shows low multicollinearity, though the “all species” figure shows 10 comparisons with a weak relationship (spearman’s >0.2), 6 of which show a strong relationship (spearman’s >0.4). Similarly, the “passerine” figure shows 8 with a weak relationship, one of which shows a strong relationship. Furthermore, you include categorical predictors that are highly correlated with your continuous predictors. I suggest you choose the best among correlated predictors (a priori) and include only them in your models. This will remove multicollinearity and make your models more parsimonious. My specific recommendations are below:

-Human disturbance: You include both a continuous and categorical estimate of human disturbance. These are certainly highly correlated (though association not provided in manuscript). Your description of categorical urban/rural determination sounds subjective, whereas the SEDAC global human footprint index sounds objective and established. I encourage you to remove the categorical urbanization predictor, and only include the continuous global human footprint index in your model.

-Life History: You include body mass, generation length, and clutch size in your model. These are all well accepted as correlated life history traits. I suggest you include only one in your model. Body mass is the most widely used of the three (as you describe in supplementary lines 47-49), and is the one you most often had species-specific data for (based off Table S1). I therefore recommend you remove generation length and clutch size, and only include body mass in your model.

-Migratory behavior: You include both categorical migratory classification and hand-wing index as independent variables. As you point out multiple times in the manuscript “hand-wing index was found to increase with migration”. From Table S1 it appears you have good data for HWI. This measure seems more objective than your categorical measure, as you don’t declare how far qualifies as “long-distance” migration. I recommend you remove the categorical migration predictor, and include only HWI in your model.

2. I would appreciate more discussion of whether you think your findings are due to evolutionary selection (bolder birds/species perform better in urban areas) vs plasticity (birds become bolder/habituated when in urban areas).

-You try to do this in lines 261-278, though I feel like this paragraph squeezes lots of important concepts together unclearly. I would make a separate paragraph where you discuss evolutionary vs. plasticity mechanisms and their associated implications.

-This distinction has important implications for both theory and management. For example, if birds in areas that are newly exposed to humans become habituated (and thus develop shorter FID) and avoid the costs you describe in lines 176-181. Therefore, if pattern arises from plasticity there would be less need for conservation buffer zones.

-One possible way to gain insight into this would be to focus on species for which you have both urban and rural observations (frequency of this should be provided in manuscript). If they show similar FID in both settings it would suggest fixed evolutionary constraints on FID and argue against plasticity.

3. Authors often used values for congeneric relatives. This should be justified.

-As presented, I'm not convinced that you shouldn't have simply omitted the 62 species shown in Table S1 (would still have robust sample of species). You chose to omit species missing other predictor values (as stated in lines 406-409). Unclear why you chose different approach for these species.

-You could take a random sample of birds that do have data and compare them to the nearest congener. If similar, that would suggest your approach is valid.

-Added benefit of omitting generation length and clutch size and instead using only body mass as life history proxy (as per Suggestion #1 above) is that you would be less reliant on values from congeneric relatives (since only 4 species seem to be missing body mass).

4. You present effect sizes, but never state the type of effect size. There are multiple possible measures of effect size. Knowing which you used is very important in determining the magnitude of the trend.

-In Figure S1 you mention Spearman's correlation coefficient. Is that also what you used for Figure 2? If so, then only start distance shows a strong relationship (>0.4). All others are negligible (0-.19). This would make me think significance is simply due to large sample sizes, not to large magnitude of effect.

-If you used a different effect size the magnitudes may be large. I just cannot tell without knowing which effect size you are reporting.

5. I am not convinced that the novelty and insight of this manuscript is suitable for Nature Communications. As I said above, I am very impressed with the scale and collaboration involved in this manuscript. However, this manuscript is basically repeating past studies (citations 29 and 55) in a new area and habitat. These original studies were published in journals with impact factors much lower than Nature Communications. I do not see anything remarkable in this manuscript that would elevate it above those original publications. I encourage authors to consider those or similar journals as a more suitable home for publication.

Minor Suggestions:

a. Change sample size in abstract to match the number you actually included in your analyses (10,249 from 842 species, listed on lines 407). These values are still very impressive. There is no reason to artificially inflate the values.

a. First paragraph of results (lines 212-225) seems like it should be moved to the methods section

b. First sentence of discussion (line 246) suggests that FID was "best predicted by" urbanization/human footprint index. However, starting distance was a far better predictor. Reword to clarify.

c. You found that tolerance was lower in larger birds. Could this be an artifact of the strong effect of starting distance since they are correlated (0.48 correlation in Figure S1). Observers would notice larger birds from farther away, and therefore observers would start approach from farther away,

and therefore birds would have larger FID. This relates back to my major comment about multicollinearity above, and also has its own implications for interpretation of body mass results.

d. Several times you describe HWI as “wing length” (e.g. line 254), though it is really wing shape since it is corrected for wing length (as stated in supplementary line 82). I would replace all mentions of “wing length” with “wing shape”.

e. Should add generation length to parenthetical statement for the passerine clade on line 254.

f. I don't feel like you adequately addressed the conflicting results of your study vs. the meta-analysis you describe in lines 300-302.

g. Lines 311-312 you say that efficient flyers may delay escape, which implies a shorter FID. However, your results show that elongated wings and migratory classification, each of which you described as “efficient” throughout the manuscript, flushed earlier. Should fix contradictory statements.

h. Your description of how you determined “sites” on lines 374-377 seems subjective. Need clear rules for when to consider it the same site (e.g. within a set distance). I'm not even convinced that you need to include site since your analysis controlled for spatial autocorrelation. Seems more appropriate to remove site variable.

i. FID of birds in flocks is highly influenced by behavior of other birds in the flock. When the first bird flies typically the rest follow. I would at least acknowledge this. Your statement on lines 395-396 suggests that the single focal bird is operating independently of the group.

j. Lines 399-400: Give numbers to support statement that the majority of data were gathered at given times (e.g. %).

k. Line 433: See the linked website for recommended citation. Should also state that you used the 2009 metric.

l. Line 448: Should provide the tree you created and used in supplementary materials.

m. Credible intervals vs. confidence intervals: You should report 95% credible intervals, not confidence intervals. I believe this is what you did since you mention credible intervals in lines 484, 815, and 820. However, in several locations you say confidence intervals (line 257, Table S2 description). Correct these to state credible intervals.

n. Line 485: You chose to report only results from $R^2 = 0.3$. Need to explain/justify this decision.

o. Figure 1: Section C shows very few urban sites. Either you sampled way more rural than urban sites (which should be stated in manuscript). Or some urban sites aren't listed on C (which seems to be the case by comparing A and C).

p. Figure 1 description: States that some urban and rural sites shared the same coordinates. How is this possible?

q. Minor grammar/wording confusion below

- Line 133 should read "...alterations are a major driver..."
- Lines 189-190: Sentence is worded improperly
- Sentence in lines 274-276 is tough to understand as worded.
- Line 517: sentence has double negative

Reviewer #3 (Remarks to the Author):

The ms reports on global variation in human tolerance among open-habitat tropical bird species, highlighting ecological and life history traits fostering human tolerance in bird populations. The authors did a huge effort to collect broad-scale data collected by different observers and to control for potentially confounding variables related to observer effects and potential heterogeneity in the dataset. However, I think a key confounding variable is missing in their analyses, i.e. a species propensity to settle in human-modified habitats. Indeed, some species would never settle in urban habitats because of their ecological or habitat requirements, for instance (thinking of open habitat species, consider several raptors or Alaudidae). Although the authors managed to control for phylogeny and life history traits, little attempt has been made to control for ecological traits (which I do understand because it is inherently difficult). Yet, my suggestion would be to perform a reanalysis of the data focusing on the subset of species which have tolerance records in both human-modified and natural habitats. Indeed, results from such an analysis should rule out any confounding effect of broader ecological requirements on the conclusions.

Finally, I miss comments to interspecies variation in cognitive abilities that can affect tolerance, of which relative brain size can be a proxy.

More detailed comments in the attached PDF file. Overall, I found this study really well conducted and I congratulate with the authors for their efforts.

**REVIEWER COMMENTS**

**PLEASE, also note that we significantly shortened abstract and a reference list to follow**
**Nature Communications guideline.**

**Reviewer #1 (Remarks to the Author):**

**This study tests the correlates of fear reactions of birds towards human observers (measured**
**as flight initiation distance, FID). The study is based on a large dataset collected by the**
**standard procedure for estimating FID. The analyses are carefully conducted and control for**
**the potential biases due to phylogenetic relatedness between species and spatial**
**autocorrelation. The results confirm most of the findings of earlier studies that used**
**(geographically) different datasets, supporting that FID are predicted by some ecological and**
**life-history variables. The study also shows that FID is a useful measure of the birds'**
**disturbance tolerance in tropical ecosystems that have been relatively rarely covered by earlier**
**studies. My comments below may help to improve some unclear points in the manuscript.**

RESPONSE: We very appreciate your comments and we did our best addressing them in an
updated version of the manuscript.

**lines 189-191: It sounds counterintuitive to say that higher predation rates / hunting pressure**
**in the tropics result in higher survival and longevity. Please provide a brief explanation.**

RESPONSE: We explained this briefly by stating that tropical birds are more risk-averse than
their temperate counterparts which may be associated with higher adult survival and longevity.
Please, note that high predation rates in tropics usually refer to nest predation. Adult predation
rates are much less known (but higher life-span in tropics is well documented). If we develop
a proxy using predation diversity normalised by species richness, predation rates may be lower
in tropics (compared to higher latitudes on Northern hemisphere) (Valcu et al. 2014,
*Ecography*, 37:930-938). Note also that survival and longevity may be affected by other factors
such as the general absence of migration in most tropical birds. (L195-197)

**I. 201-203: A longer FID indicates LESS tolerance to human disturbance while a shorter FID**
**indicates HIGHER tolerance (i.e. the opposite of what is claimed here).**

RESPONSE: Many thanks for your careful reading! Of course, you are right here – corrected.
(L208-210)

**I. 221, further information on data sources in Table S1: Body mass data lacking for a species**
**were substituted by body mass from a closely related species (Supplementary Methods I. 53-**
**54). Then, for some species that also lack clutch size data this was substituted by data "from**
**congeneric species with relatively similar body sizes" (Supplementary Methods I. 65-66). Does**

this means that both body mass and clutch size were used from the same "donor" species? If
yes, please give this information explicitly somewhere, e.g. in the Supplementary Methods.

RESPONSE: Thank you for this comment. In the current version of manuscript, we decided to
not impute values from closely related species.

45 l. 263-264: "... based on their syndromes" -> do you mean behavioural syndromes?

RESPONSE: Yes, indeed. We added word "behavioural" here. (L286)

48 l. 380-381: "All escape data were collected using standard procedure developed by ref.38." -
49 Ref. 38 is from 2006, while data collection started in 2000 in some study sites (see l. 361-370)
- thus not all data collection could follow the procedure published in 2006.

RESPONSE: All data collected before 2006 were collected by Daniel Blumstein who
developed this method and first published in 2003 and 2006. Hence, also data before 2006
were collected using this procedure although at that time it was still unpublished. In a current
version of manuscript, we cited here also older paper on this topic from 2003 and deleted
"developed by" to avoid any confusion. (L438)

57 l. 396: "Researchers wore standardised outdoor clothing" - what do you mean on standardized,
e.g. all observers' clothing had the same colour?

RESPONSE: We deleted "standardised" here to better capture the real situation. (L454)

61 l. 437-438: "urban: areas with continuous urban elements like multi-storey buildings, family
houses or roads" - What was the basis of the assessment that the sites contain these urban
elements or not? Please clarify whether this was based on information from specific databases
or maps.

RESPONSE: All habitat assessments (i.e. whether sampled site contained human-made
elements) were done during the fieldwork. We now mention this info explicitly here. (L495-502)

68 l. 316-331, explanations for the seasonal change in FID: An important alternative explanation
is that the bird communities of the studied areas change seasonally, e.g. due to the
presence/absence of migrants, that may affect FID. Although both the migration status of the
species and the season of the observations were included as predictors in the multi-predictor
models, their effects may be difficult to separate due to their potential collinearity. Although the
authors claim that multicollinearity is low among predictors (l. 470-472), this is only shown for
continuous predictors (Fig S1), thus it is unclear whether season and migration status was
strongly correlated (as can be assumed by the seasonal nature of migration) or not. To make
the independent effect of season more convincing, I suggest to demonstrate that these
predictors are not/weakly correlated. A brief discussion of the potential confounding effect of
migration (especially if that is justified by its strong correlation with season) would also be

**useful.**

RESPONSE: Many thanks for these comments. We updated collinearity matrix in SI, showing
no strong correlation between season and migration (i.e., $r = -0.13$ and -0.15 in full and
passerine dataset, respectively) (Fig S1). Hence, we should be able to separate effects of both
variables in our analyses.

Reviewer #2 (Remarks to the Author):

This study aims to assess predictors of flight initiation distance among birds in open tropical
systems, with a particular emphasis on urbanization. The authors used data collection across
an enormous geographic scale and species breadth. I commend the authors for their creativity
and cooperation to execute such an ambitious project. Researchers often operate and publish
within their respective silos; it is refreshing to see researchers from across the world combine
data into a collaborative manuscript.

The manuscript is mostly well-written and clear, the research question is valid, and the
methodologies appear consistent across researchers. However, I do not believe it is
publishable in its current form due to the major suggestions listed below. I do believe it is
possible to revise the manuscript and successfully publish it. I hope the authors take the
suggestions below as constructive and use them to improve and prepare this manuscript for
ultimate publication.

RESPONSE: Many thanks for your kind and supportive words. We very appreciate your
comments and did our best to address them in this revised manuscript.

Major Suggestions:

1. The analyses simultaneously include correlated independent variables. This multicollinearity
threatens the validity of your analyses. You say that Figure S1 shows low multicollinearity,
though the “all species” figure shows 10 comparisons with a weak relationship (spearman’s
>0.2), 6 of which show a strong relationship (spearman’s >0.4). Similarly, the “passerine” figure
shows 8 with a weak relationship, one of which shows a strong relationship. Furthermore, you
include categorical predictors that are highly correlated with your continuous predictors. I
suggest you choose the best among correlated predictors (a priori) and include only them in
your models. This will remove multicollinearity and make your models more parsimonious. My
specific recommendations are below:

RESPONSE: We now provide correlation matrix for both continuous and categorical predictors
(Fig S1). We clearly show that collinearity between predictors is usually low and never above
0.7 (which is a commonly used threshold for omitting one or more correlated predictors from
the analysis). All correlation coefficients were <0.5 with one exception – correlation between
habitat type and human footprint ($r = 0.55$ and 0.62 in full and passerine dataset, respectively).
To exclude the possibility that this correlation affected our results, we now provide two
alternative model, either with habitat type and human footprint included. We found that in these
models, both habitat type and human footprint were significant predictors of escape behaviour
of birds (see method and results section).

-Human disturbance: You include both a continuous and categorical estimate of human
disturbance. These are certainly highly correlated (though association not provided in
manuscript). Your description of categorical urban/rural determination sounds subjective,
whereas the SEDAC global human footprint index sounds objective and established. I
encourage you to remove the categorical urbanization predictor, and only include the
continuous global human footprint index in your model.

RESPONSE: Thank you for this comment. We fully acknowledge that habitat type (urban/rural)
is more subjective than human footprint index. However, we would like to use both these
indexes because habitat type in terms of urbanization is long-established and widely-used
proxy for the level of human disturbance in FID studies (e.g. Díaz et al. 2013, PLoS one
8:e64634; Carrete et al. 2016, Sci Rep 6:1-6; Rebolo-Ifrán et al. 2015, Sci Rep 5:1-10). We
added these reasoning also to the main text (methods and discussion; e.g. L495-502) and SI.
Moreover, the fact that the effects of both predictors are supported together in the full model
suggests they contain information that is at least partly specific.

-Life History: You include body mass, generation length, and clutch size in your model. These
are all well accepted as correlated life history traits. I suggest you include only one in your
model. Body mass is the most widely used of the three (as you describe in supplementary lines
47-49), and is the one you most often had species-specific data for (based off Table S1). I
therefore recommend you remove generation length and clutch size, and only include body
mass in your model.

RESPONSE: In this revision, we omitted generation length from all analyses. However, we
would like to keep body mass and clutch size in our analyses because clutch size can be used
as a proxy for investments to reproduction (species with similar body masses may markedly
differ in their reproductive strategies and investments) whereas body mass is correlated with
many other important life-history traits and determines the overall position along slow-fast
pace-of-life continuum in animals. Because we were interested in separating the effects of
body mass and clutch size on FID and control for their mutual effects (i.e., the effect of body
mass independent of clutch size and the effect of clutch size independent of body mass,
respectively), we must include both predictors in the same model. Importantly, there is no
strong collinearity since the correlation between body mass and clutch size is generally very
low ($r < 0.2$) (Fig S1).

-Migratory behavior: You include both categorical migratory classification and hand-wing index
as independent variables. As you point out multiple times in the manuscript “hand-wing index
was found to increase with migration”. From Table S1 it appears you have good data for HWI.
This measure seems more objective than your categorical measure, as you don't declare how
far qualifies as “long-distance” migration. I recommend you remove the categorical migration
predictor, and include only HWI in your model.

RESPONSE: Many thanks for this interesting comment. If possible, we would like to keep both
migration and HWI in our models. Migration was used as a proxy of long-distance movements
of birds whose breeding and wintering ranges do not overlap, while HWI may serve as a proxy
of overall flight ability and performance (and is linked not only to migration but also style of
food acquisition, aerial displays, and general lifestyle). Hence, the association between each
of these predictors and FID should be interpreted differently. For example, besides indicating
flight ability, migration can be a metric of the familiarity of birds with tropical areas; tropical
residents spend all year in these areas whereas temperate zone migrants spend only part of
the year in this region. The assumption that both indices capture to different lifestyle
characteristics is further supported by the relatively low correlation between the two variables
($r = 0.49$ and 0.38 in full and passerine dataset, respectively) (Fig S1).

In the Supplementary Methods, we at least attempted to clarify how the different migratory
categories were defined. For example, a species was considered a long-distance migrant if “a
substantial proportion of the global or regional population makes regular / seasonal cyclical
movements beyond the breeding range”.

2. I would appreciate more discussion of whether you think your findings are due to
evolutionary selection (bolder birds/species perform better in urban areas) vs plasticity (birds
become bolder/habituated when in urban areas).
-You try to do this in lines 261-278, though I feel like this paragraph squeezes lots of important
concepts together unclearly. I would make a separate paragraph where you discuss
evolutionary vs. plasticity mechanisms and their associated implications.
-This distinction has important implications for both theory and management. For example, if
birds in areas that are newly exposed to humans become habituated (and thus develop shorter
FID) and avoid the costs you describe in lines 176-181. Therefore, if pattern arises from
plasticity there would be less need for conservation buffer zones.

RESPONSE: Thank you for this comment. We have now expanded this section. However,
even after additional analysis of birds approached in both rural and urban habitats (see also
the comment below), which supported the within-species mechanisms, we cannot clearly
distinguish between three conceivable mechanisms. First, it is possible that some individuals
that have inherently high tolerance towards humans and exhibit personality-dependent habitat
preference. Second, some birds may be in general behaviourally very plastic and increase
their tolerance towards human when necessary. Third, selection against risk-averse individuals
may be present in urban habitats. Moreover, we still cannot exclude some species-level
mechanisms affecting avian tolerance towards humans, such as a higher probability of urban
habitats to be colonised by risk-tolerant species.

-One possible way to gain insight into this would be to focus on species for which you have
both urban and rural observations (frequency of this should be provided in manuscript). If they
show similar FID in both settings it would suggest fixed evolutionary constraints on FID and
argue against plasticity.

RESPONSE: In this revision, we fitted models using a subset of species with FIDs measured
in both urban and rural habitats. We found that populations birds differed in their FIDs between
rural and urban habitats both in all species and in passerines, respectively. This indicates that
either behavioural plasticity, preferential urban colonization by the rural individuals which are
already bold or selection against risk-averse individuals in urban environments are important
mechanism behind tolerance towards humans in animals. Again, it's not straightforward to infer
mechanism from such correlative results. Please, see our updated discussion.

3. Authors often used values for congeneric relatives. This should be justified.

-As presented, I'm not convinced that you shouldn't have simply omitted the 62 species shown
in Table S1 (would still have robust sample of species). You chose to omit species missing
other predictor values (as stated in lines 406-409). Unclear why you chose different approach
for these species.

-You could take a random sample of birds that do have data and compare them to the nearest
congener. If similar, that would suggest your approach is valid.

-Added benefit of omitting generation length and clutch size and instead using only body mass
as life history proxy (as per Suggestion #1 above) is that you would be less reliant on values
from congeneric relatives (since only 4 species seem to be missing body mass).

RESPONSE: In the revision, we elected to not impute missing values from closely related
species. Despite a slightly smaller sample size, the main results remained qualitatively the
same.

4. You present effect sizes, but never state the type of effect size. There are multiple possible
measures of effect size. Knowing which you used is very important in determining the
magnitude of the trend.

RESPONSE: We report standardized effect size in the form of standardized partial regression
coefficients (Schieletz 2010, Methods Ecol Evol 1:103-113) obtained by dividing the
continuous variables by two standard deviations (Gelman 2008, Stat Med 27:2865-2873). The
advantage of standardizing the continuous variables by two standard deviations is that the
resulting effects of continuous predictors are directly comparable to those of categorical
predictors. The interpretation of standardized partial regression coefficient is simple: an effect
size of 1 means that a change in the predictor by one standard deviation is equal to a change
in the dependent variable by one standard deviation while all the other predictors are kept
constant (at their mean values). We now clearly provide this information in the main text (see
method section).

-In Figure S1 you mention spearman's correlation coefficient. Is that also what you used for
Figure 2? If so, then only start distance shows a strong relationship (>0.4). All others are
negligible (0-.19). This would make me think significance is simply due to large sample sizes,
not to large magnitude of effect.

RESPONSE: No. Figure S1 shows correlation between predictors (FID as response variable
is not included) – the aim of this figure is to assess potential multicollinearity between
predictors. Figure 2 shows the main results – standardised effect sizes of the predictors (while
controlling for their mutual effects) on FID based on phylogenetically and spatially informed
Bayesian mixed models. For each predictor, we report standardised effect size (see above) in
Fig. 2. For example, effects <0.2 are relatively weak, meaning that a change in the predictor
by one standard deviation equal to a change in the dependent variable by 0.2 of its standard
deviation. The two figures are independent in their meaning.

-If you used a different effect size the magnitudes may be large. I just cannot tell without
knowing which effect size you are reporting.

RESPONSE: Please, see our comments above.

5. I am not convinced that the novelty and insight of this manuscript is suitable for Nature

Communications. As I said above, I am very impressed with the scale and collaboration
involved in this manuscript. However, this manuscript is basically repeating past studies
(citations 29 and 55) in a new area and habitat. These original studies were published in
journals with impact factors much lower than Nature Communications. I do not see anything
remarkable in this manuscript that would elevate it above those original publications. I
encourage authors to consider those or similar journals as a more suitable home for
publication.

RESPONSE: We would like to highlight that our study, in contrast to many of previous studies:
(a) focuses on tropical regions that are still largely understudied. In our study, we put together
a diverse team of collaborators and put together large amount of data (most of which has never
been published before). Hence, our study may serve as a benchmark for future studies on
animal risk-taking in tropical ecosystems; (b) we make all primary data publicly available
(previous studies made typically available only species or population means, often without any
additional descriptor variables); and (c) some of associations found are rarely studied and
documented in the literature (e.g. association between FID and migration, wing shape, and
clutch size, respectively).

We think that the studies on animal fear of human and escape behaviour in wild-living animals
is a rapidly growing field. Large-scale papers on this topic are published in high-IF journals and
are often highly cited (e.g. Samia et al. 2015, Nat Comm 6:1-8; Morelli et al. 2018, Sci Total
Environ 631:803-810; Morelli et al. 2022 Sci Total Environ 160534). We also believe that
creating a scientific culture focused on replication in high-impact multidisciplinary/ecology
journals may partially mitigate the replication crisis in ecological research (e.g. Filazzola &
Cahill 2021, Methods Ecol Evol 12:1780-1792; Makel & Plucker 2014, Educ Res, 43:304-316).

Minor Suggestions:
a. Change sample size in abstract to match the number you actually included in your analyses
(10,249 from 842 species, listed on lines 407). These values are still very impressive. There is
no reason to artificially inflate the values.

RESPONSE: Thank you. We changed this.

a. First paragraph of results (lines 212-225) seems like it should be moved to the methods
section

RESPONSE: Thank you for this comment. If possible, we would like to keep the text here in a
present form. When writing particular sections of the manuscript, we were inspired by the style
of other paper published in Nat Comm – many papers used a brief Method introduction at the
beginning of Results section (because main method section is located after Results section in
Nat Comm).

b. First sentence of discussion (line 246) suggests that FID was “best predicted by”
urbanization/human footprint index. However, starting distance was a far better predictor.
Reword to clarify.

RESPONSE: In the present manuscript, we first mention starting distance and only then the
effect of urbanization/human footprint index. (L262-263)

c. You found that tolerance was lower in larger birds. Could this be an artifact of the strong
effect of starting distance since they are correlated (0.48 correlation in Figure S1). Observers
would notice larger birds from farther away, and therefore observers would start approach from
farther away, and therefore birds would have larger FID. This relates back to my major
comment about multicollinearity above, and also has its own implications for interpretation of
body mass results.

RESPONSE: We controlled escape responses of birds for starting distances of approaching
human in the statistical analyses. Hence, even if larger birds would be approached from longer
distances, this effect should be statistically controlled for. In fact, the effect of body mass could
be confounded by its correlation with starting distance, only if starting distance was excluded
from the model. In contrast, when both collinear predictors are included in the model, the part
of explained variance in FID that is shared by both predictors is excluded from the estimation
of their effects. Hence, including both collinear predictors ensures that the estimated effects
are not due to the collinearity but are instead specific and independent of the effect of the other
predictor (please see Freckleton 2002, J Anim Ecol 70:708-711).

317 d. Several times you describe HWI as “wing length” (e.g. line 254), though it is really wing
shape since it is corrected for wing length (as stated in supplementary line 82). I would replace
all mentions of “wing length” with “wing shape”.

RESPONSE: Thank you, corrected.

e. Should add generation length to parenthetical statement for the passerine clade on line 254.

RESPONSE: We excluded generation length from the analysis, following comments above.

f. I don't feel like you adequately addressed the conflicting results of your study vs. the meta-
analysis you describe in lines 300-302.

RESPONSE: We deleted this sentence because it was related to our results only indirectly.
We found that larger birds escape earlier than smaller birds which is in a good agreement with
the results of Samia et al. meta-analysis. In contrast to this meta-analysis, we were not
interested in whether larger birds exhibited more pronounced reduction in their escape
distances than smaller birds as human disturbance increased.

333 g. Lines 311-312 you say that efficient flyers may delay escape, which implies a shorter FID.
However, your results show that elongated wings and migratory classification, each of which
you described as “efficient” throughout the manuscript, flushed earlier. Should fix contradictory
statements.

RESPONSE: Thank you for careful reading! We corrected this mistake.

339 h. Your description of how you determined “sites” on lines 374-377 seems subjective. Need
clear rules for when to consider it the same site (e.g. within a set distance). I’m not even
convinced that you need to include site since your analysis controlled for spatial
autocorrelation. Seems more appropriate to remove site variable.

RESPONSE: We now fully acknowledge in the text that our approach brings some level of
subjectivity (429-434). We were not able to apply the distance criterion on our data because
some data were collected during long-distance transect studies. In such cases, “one locality”
would cover several tens of kilometres. We believe that our current approach is more
biologically relevant and meaningful than strict application of distance rule. Site as well as
species need to be included in models even when spatial and phylogenetic matrices,
respectively, are included in models to control for intraspecific variation in traits.

i. FID of birds in flocks is highly influenced by behavior of other birds in the flock. When the
first bird flies typically the rest follow. I would at least acknowledge this. Your statement on
lines 395-396 suggests that the single focal bird is operating independently of the group.

RESPONSE: We now clearly acknowledge this in the text. (L452-454)

j. Lines 399-400: Give numbers to support statement that the majority of data were gathered
at given times (e.g. %).

RESPONSE: This information was added. (L458)

k. Line 433: See the linked website for recommended citation. Should also state that you used
the 2009 metric.

RESPONSE: We added recommended citations and stated that it was 2009 metric. (L490-
491)

365 l. Line 448: Should provide the tree you created and used in supplementary materials.
366 m. Credible intervals vs. confidence intervals: You should report 95% credible intervals, not
confidence intervals. I believe this is what you did since you mention credible intervals in lines
484, 815, and 820. However, in several locations you say confidence intervals (line 257, Table
S2 description). Correct these to state credible intervals.

RESPONSE: All data, tree and code are now freely available and attached to the manuscript
(<https://doi.org/10.17605/OSF.IO/BSPQX>). Confidence intervals are usually labelled as
credible intervals in Bayesian statistics (e.g. see documentation to brms package that we used
for regression analyses <https://cran.r-project.org/web/packages/brms/index.html>). Hence, we
decided to use term “credible interval” throughout the text.

n. Line 485: You chose to report only results from $R^2 = 0.3$. Need to explain/justify this decision.

RESPONSE: The main purpose of the prior (for predictor effects) that is scaled by the number
of predictors and the prior belief about R2 is to prevent overfitting. The scaling results in a prior
that concentrates more probability around zero and provides less prior probability for larger
effects. It should be noted that scaling is often done by dividing the expected R2 value with the
number of predictors and a prior with any R2 value would still be relatively conservative
compared to unscaled priors that are commonly used. We have chosen prior belief $R^2 = 0.3$
as this seems to be a reasonable proportion of variance that could be explained by the model,
given that the model included several predictors previously shown to be associated with FID
(L550-554). Moreover, the sensitivity analysis showed that the results are robust to the effect
of prior R2, with all models providing the qualitatively same results for all predictors (see Table
S1).

o. Figure 1: Section C shows very few urban sites. Either you sampled way more rural than
urban sites (which should be stated in manuscript). Or some urban sites aren't listed on C
(which seems to be the case by comparing A and C).

RESPONSE: Yes, we sampled less urban than rural sites. As we mentioned in the figure
caption, "Note that some very nearby urban and rural sites shared the same geographic
coordinates – for clarity, these sites were excluded from panels b, and c, respectively."

p. Figure 1 description: States that some urban and rural sites shared the same coordinates.
How is this possible?

RESPONSE: Sometimes, data contributors entered adjacent urban and rural areas with the
same geo-coordinates. This should have no significant impact on results and their
interpretation because: (a) we clearly distinguished between urban and rural areas in the
analyses, and (b) we used geo-coordinates only to calculate distance matrix with nearby
localities having shorter distances between each other than more distant localities; our dataset
is really large scale, covering localities across thousands of kilometres and treating some of
closely positioned localities as having zero distance (instead a very small distance of hundreds
meters or very few kilometres) between each other should have a minimal effect on the overall
general patterns and findings.

q. Minor grammar/wording confusion below
-Line 133 should read "...alterations are a major driver..."

RESPONSE: Corrected.

-Lines 189-190: Sentence is worded improperly

RESPONSE: We tried to reword this sentence and make it clearer.

-Sentence in lines 274-276 is tough to understand as worded.

RESPONSE: We reworded this sentence.

-Line 517: sentence has double negative

RESPONSE: Corrected.

Reviewer #3 (Remarks to the Author):

The ms reports on global variation in human tolerance among open-habitat tropical bird
species, highlighting ecological and life history traits fostering human tolerance in bird
populations. The authors did a huge effort to collect broad-scale data collected by different
observers and to control for potentially confounding variables related to observer effects and
potential heterogeneity in the dataset. However, I think a key confounding variable is missing
in their analyses, i.e. a species propensity to settle in human-modified habitats. Indeed, some
species would never settle in urban habitats because of their ecological or habitat
requirements, for instance (thinking of open habitat species, consider several raptors or
Alaudidae). Although the authors managed to control for phylogeny and life history traits, little
attempt has been made to control for ecological traits (which I do understand because it is
inherently difficult). Yet, my suggestion would be to perform a reanalysis of the data focusing
on the subset of species which have tolerance records in both human-modified and natural
habitats. Indeed, results from such an analysis should rule out any confounding effect of
broader ecological requirements on the conclusions.

RESPONSE: Many thanks for this comment. In all analyses, we controlled for confounding
effect of habitat type in terms of urbanization. Habitat type was scored for each bird population
as a binary variable (rural or urban). Hence, we had very precise information on which birds
were sampled in rural habitats or in the vicinity of human settlements and we controlled for this
factor in our analyses. In this revised version of the manuscript, we also fitted models using a
subset of species with FIDs measured in both urban and rural habitats. We still found
differences in avian tolerance towards humans between urban and rural habitats. However,
we would like to note that even this analysis cannot rule out the species-level mechanisms
which can still be potentially involved (see our other comments and discussion in the main
text).

Finally, I miss comments to interspecies variation in cognitive abilities that can affect tolerance,
of which relative brain size can be a proxy.

RESPONSE: Many thanks for this comment. We avoided including brain size in our models
because: (1) brain size is very strongly correlated with body size (Sayol et al. 2016, Nat Comm
7:1-8), (2) brain size is often missing for tropical species. Even the largest available datasets
for brain size covers <10% of all passerine species (e.g. reference above) with most species
breeding in temperate regions, and (3) although relative brain size is widely used in
comparative studies, FIDs seem to be correlated more closely with the size of specific brain
compartments than whole brain size (Symonds et al. 2014, PLoS One 9:e91960). We fully
agree that this is a very interesting topic but we feel that this topic is very complex and requires
a separate in-depth and well-designed study. Thus, and since this topic is beyond the scope
of this study, we would prefer to not speculate more in this current study.

More detailed comments in the attached PDF file. Overall, I found this study really well
conducted and I congratulate with the authors for their efforts.

RESPONSE: Many thanks for your very supportive words!

Comments from PDF

L189-190: Not sure whether 'causing' is the correct word here, in evolutionary terms. Perhaps
this sentence is too strong.

RESPONSE: We have rewritten this part.

L201-203: This sentence is confusing: I would say that birds which allows a closer approach
by humans is more (not less!) tolerant of human disturbance. Please check out also with
Discussion: Avian tolerance towards humans decreased (i.e. escape distance increased)

RESPONSE: Many thanks for your careful reading. You are right here – corrected. (L208-210)

L212: replace “for” by “at”

RESPONSE: Done.

L218-219: This is a bit complicate of course. How does this covary with measures of human
disturbance of habitats? I mean, if starting distances were shorter in more disturbed compared
to less disturbed habitats? I think this is an important potential confound of the patterns of
response in relation to human pressure which may lead to spurious findings, so potential
collinearity effects should be evaluated.

RESPONSE: Thank you. We provide correlations between starting distance and all other
predictors in Fig. S1. We generally found low collinearity between starting distance and habitat
type ($r = -0.21$ and -0.23 in the full dataset and in the passerine dataset, respectively). Most
importantly, we included both predictors in the models, so their mutual effects are controlled
for. Both starting distance and habitat are significant predictors, so there should be no problem.
Collinearity does not create non-existent effects or bias their estimates. Collinearity can only
increase uncertainty about the estimates (i.e., widen the credible/confidence intervals). Please,
see also Freckleton (2002; J Anim Ecol 70:708-711).

L221: A potential key variable to explain interspecific variation in human disturbance that I am
missing here is the tendency to urbanization of a species. Not all species are equally prone to
settle in urban habitats, because of different lifestyle or ecological requirements. And this may
or may not depend on their human tolerance. Perhaps an indicator of the overall tendency to
frequent urban habitats should be included (but this would require a huge additional work). But
perhaps, you can consider running your analyses on the subset of species which have been
observed in both disturbed and undisturbed habitats. The results may reveal factors promoting
tolerance irrespective of confounding effects of ecological differences between species, which
can only partly be controlled for by phylogenetic analyses.

RESPONSE: Many thanks for this comment. In this revision we also fitted models using a
subset of species with FIDs measured in both urban and rural habitats. We found that
populations birds differed in their FIDs between rural and urban habitats both in all species and
in passerines, respectively. This indicates that either behavioural plasticity, preferential urban
colonization by the rural individuals which are already bold or selection against risk-averse
individuals in urban environments are important mechanism behind tolerance towards humans
in animals. Again, it's not straightforward to infer mechanism from such correlative results.
Please, see our updated discussion.

L228-229: please check coherence with Introduction regarding 'tolerance'. Also, 'sooner' can
be a bit misleading here as you estimated the escape 'distance', not 'timing'. Please consider
rewording

RESPONSE: Many thanks. We checked this and corrected in the introduction. We replaced
"birds escaped sooner from an approaching human" by "escape distance was longer".

L264: behavioural syndromes

RESPONSE: Added.

L271: potentially important

RESPONSE: Added.

L317-319: of course there are many different possible alternatives, including: seasonal
variation in age structure of the population, whereby you may expect more juveniles (typically
more tolerant) than adults (less tolerant) in the dry vs. wet season, or different stress
susceptibility levels related to seasonal changes in resource abundance.

RESPONSE: We tried to downplay a bit this explanation by replacing "indicate" by "suggest"
(L357). We mentioned many alternative explanations in the last sentence of this paragraph
(including your suggestion "different stress susceptibility levels related to seasonal changes in
resource abundance") – we included there also your suggestion related to seasonal
differences in age structure of birds.

L327-328: This finding is also somewhat counterintuitive, although without individual-level data
it is difficult to say with certainty what is going on. In any case, stress levels are well-known to
trigger dispersal decisions (see early studies on dispersal of American kestrels by G. Bortolotti
and coworkers). Perhaps you can add breeding territoriality as a further variable - then I can
envisage an ecological pathway such as: breeding territoriality -> higher intra/interspecific
competition -> higher stress (CORT) levels -> greater dispersal propensity -> lower tolerance

RESPONSE: Many thanks for this interesting comment. Unfortunately, detailed data on
territoriality for many tropical species are still rare, assuming that most tropical species are
territorial year-round or almost year-round. We partly controlled for the effect of territoriality by
including migration as a covariate in the analyses which clearly divide species in mostly
temperate species with short-term territoriality and mostly tropical residents exhibiting long-
term territoriality. As this is rarely studied topic and we feel that we cannot address it properly
at this stage, and would prefer to not speculate more on this in the current manuscript.

L365-371: As data were collected over a long period, temporal trends should be assessed.

RESPONSE: We re-ran our analyses controlling also for the effect of year (included as a
random factor) (L526). We found that the main results remain qualitatively unchanged. We
included this information in the method section of the manuscript.

L373-374: This casts some doubts on the extent of data collection standardization. Indeed,
different observers may have adopted different data collection protocols, and different
interpretation of FID. Please check for observer effects, which may bias your findings.

RESPONSE: The collector effect was included and controlled in our original regression models
(L527).

L393-394: This is of course rather subjective, which increase the risk for observer effects.

RESPONSE: We added word “considerable” before “signs of distress” (L450). As mentioned
above, the collector effect was included and controlled in our original regression models.

L394: This is also something that cannot be controlled for because the location of nests was
unknown. Please avoid or rephrase

RESPONSE: We rephrased this into “Observers did not approach birds at their nests.”. (L451-
452)

L446-447: 100 trees is quite a limited sample of posterior trees to build a consensus tree.
Previous empirical evaluations suggested that 1000 trees could improve reliability of parameter
estimates in phylogenetic analyses (see Rubolini et al. 2015)

RESPONSE: In a current version of manuscript, we build MCC tree using 1000 trees. (L507-
509)

L456: do you mean the 'phylogenetic distance' in the phylogenetic distance matrix? 'Patristic
distance' sounds odd to me...

RESPONSE: We changed this but we would like to point that “patristic distance” is commonly
used term (e.g. Prosperi et al. 2011, Nat Comm 2:321; Gittleman et al. 1990, Syst Zool 39:227-
241).

L458-459: This is of course 'phylogenetic' time.

RESPONSE: We changed this but we would like to point that “patristic distance” is commonly
used term (e.g. Prosperi et al. 2011, Nat Comm 2:321; Gittleman et al. 1990, Syst Zool 39:227-
241).

L465-466: This solves (at least partly) my previous comments on observer effects. Perhaps
you can mention it earlier in the manuscript. I wonder whether site and species should be
included as further random intercept effects, besides collector ID.

RESPONSE: Species (resp. phylogeny) as well as site (resp. distance matrix) were included
as random intercepts and covariance matrices in our original models. We made this information
clear in the modelling description. (L524-527)

L471: report how low explicitly

RESPONSE: All values are explicitly stated in Figure S1.

REVIEWERS' COMMENTS

Reviewer #2 (Remarks to the Author):

The authors seemed to carefully consider and address suggestions made by myself (Reviewer #2) and the other reviewer. I commend the authors for their thorough responses and substantial revisions. I believe these changes improved the clarity of the manuscript and will make it more compelling to readers. I have no further suggestions and am convinced that the manuscript is now suitable for acceptance. Congratulations to the authors on a very ambitious and interesting study!

-Doug Barron

Reviewer #3 (Remarks to the Author):

Overall, I think the authors did a good job in revising the manuscript and have provided convincing arguments in their response letter. Yet, I have some further issues I would like the authors to address.

First, are you sure your title is grammatically correct? I think 'the' should be removed. Moreover, the meaning of 'disturbed' would not be immediately clear to most (general) readers of Nat Comm. Moreover, I think it is to some extent misleading. How can you say that 'urban' habitats are 'open'? They are not, of course (you do not have savannah in urban habitats, you have buildings, which largely generate impervious and relatively 'closed' habitats). Both rural and urban habitats are heavily man-modified habitats (rural is mostly used by referring to pastoral or agricultural areas, to my understanding), so the rural-urban gradient basically reflects a gradient of increasing density of human settlements in heavily human altered landscapes (either by agriculture or by urbanization), whereas wildlands are excluded. Were observations conducted also in wild areas? What is the spread of human footprint in your 'urban' and 'rural' sampling sites? I think a histogram showing the distribution of human footprint of urban and rural sites (in supplementary material) would be very useful. The correlation between habitat type and human footprint is indeed relatively low (ca. 0.5), reflecting that fact that both urban and rural sites show a high variance in human footprint. In any case, while you define (quite accurately) what 'urban' habitats are, you do not define at all what 'rural' habitats are. Perhaps you should spend a few words to better characterize 'rural' sites, by clarifying if natural areas are considered or if you only included agricultural areas surrounding urban sites. Back to the title, I suggest rewording with a more straightforward alternative, such as: Birds show increased human tolerance in man-modified tropical habitats

Other comments:

L249: In supplementary Table 1 I cannot find the details of this analysis. I also suggest including estimates in Figure 2 (it would just be a further bar per variable).

L269: sentence is repetitive and confusing (but..but...), please reword.

L296-298: I do agree with this reasoning, but the sentence is difficult to follow and needs reworking. I.e. "Altogether, the results of the analysis restricted to the same-species and populations occurring in both rural and urban habitats are consistent with mechanisms acting at the intraspecific level." would be way more straightforward.

L298: I would avoid "Unfortunately, however,...", use either one or the other

L309: "studies work to distinguish" ...odd wording, rephrase

L370-383: this paragraph is overlong considering that you state this effect is 'weak' and not confirmed in other subsets of the analyses.

L384: of course, there is always some association, so strictly speaking this sentence is wrong (no association). You may say that you found very weak, negligible, or non-significant associations, but you should never say that you found no association.

L397-410: I think this last paragraph does not properly convey the key messages of the study. It is mostly focused on 'applied' issues, whereas this is by far an evolutionary ecology study, investigating broad scale patterns of adaptation of wild species to human disturbance and tolerance of disturbed habitats. That this study was developed "in order to help developing global, evidence-based management interventions" seems out of scope to me, besides being unclear which kind of evidence-based management interventions can be hypothesized based on these results. Perhaps you may explain a bit more (i.e. you may suggest that in an increasingly human dominated world conservation efforts should be more focused on less tolerant species?). In any case, I think you should focus more on the factors promoting adaptation (or lack of adaptation) and its ecological/evolutionary consequences.

In general, I think the discussion is excessively speculative and not focused on the main findings. I think it can be considerably reduced by stating from the beginning that it will focus on the more robust pattern (i.e. those confirmed across multiple datasets).

REVIEWERS' COMMENTS

Reviewer #2 (Remarks to the Author):

The authors seemed to carefully consider and address suggestions made by myself (Reviewer #2) and the other reviewer. I commend the authors for their thorough responses and substantial revisions. I believe these changes improved the clarity of the manuscript and will make it more compelling to readers. I have no further suggestions and am convinced that the manuscript is now suitable for acceptance. Congratulations to the authors on a very ambitious and interesting study!

-Doug Barron

Many thanks for supportive words! We very much appreciated your comments.

Reviewer #3 (Remarks to the Author):

Overall, I think the authors did a good job in revising the manuscript and have provided convincing arguments in their response letter. Yet, I have some further issues I would like the authors to address.

Many thanks for supportive words! We did our best in addressing your comments.

First, are you sure your title is grammatically correct? I think 'the' should be removed. Moreover, the meaning of 'disturbed' would not be immediately clear to most (general) readers of Nat Comm. Moreover, I think it is to some extent misleading. How can you say that 'urban' habitats are 'open'? They are not, of course (you do not have savannah in urban habitats, you have buildings, which largely generate impervious and relatively 'closed' habitats).

Thank you for your comment. Following your and Senior Editor suggestions, we changed the title of our manuscript to more general "Bird tolerance to humans in open tropical ecosystems". However, we think that most of urban areas (at least in our samples) can be considered as "open" habitats. For example, most our data were collected in urban parks which are often relatively open in terms of tree vegetation and may resemble savannah ecosystems. Moreover, natural rocky habitats (which resemble built-up urban areas) without continuous tree cover are usually considered as open habitats.

Both rural and urban habitats are heavily man-modified habitats (rural is mostly used by referring to pastoral or agricultural areas, to my understanding), so the rural-urban gradient basically reflects a gradient of increasing density of human settlements in heavily human altered landscapes (either by agriculture or by urbanization), whereas wildlands are excluded. Were observations conducted also in wild areas?

Thank you, this is good comment. In the methods section, we now clearly state that rural areas included both natural and agricultural landscapes.

What is the spread of human footprint in your 'urban' and 'rural' sampling sites? I think a histogram showing the distribution of human footprint of urban and rural sites (in supplementary material) would be very useful. The correlation between habitat type and

human footprint is indeed relatively low (ca. 0.5), reflecting that fact that both urban and rural sites show a high variance in human footprint.

Thank you, we now provide histogram showing the distribution of human footprint of urban and rural sites in supplementary material.

In any case, while you define (quite accurately) what 'urban' habitats are, you do not define at all what 'rural' habitats are. Perhaps you should spend a few words to better characterize 'rural' sites, by clarifying if natural areas are considered or if you only included agricultural areas surrounding urban sites.

In the methods section, we now clearly state that rural areas included both natural and agricultural landscapes.

Back to the title, I suggest rewording with a more straightforward alternative, such as: Birds show increased human tolerance in man-modified tropical habitats

Thank you for your comment. Following your and Senior Editor suggestions, we changed the title of our manuscript to more general "Bird tolerance to humans in open tropical ecosystems".

Other comments:

L249: In supplementary Table 1 I cannot find the details of this analysis. I also suggest including estimates in Figure 2 (it would just be a further bar per variable).

In Table 1 caption, we added a statement where details on statistical analyses can be found ("For details on statistical analyses, see method section in the main text."). We decided not to add extra bar in the forest plot because (a) it could decrease readability of the forest plot, and (b) results of all analyses shows the same results regarding the main study questions. However, we tried to make our supplementary tables more clear.

L269: sentence is repetitive and confusing (but..but...), please reword.

We reworded this sentence.

L296-298: I do agree with this reasoning, but the sentence is difficult to follow and needs reworking. I.e. "Altogether, the results of the analysis restricted to the same-species and populations occurring in both rural and urban habitats are consistent with mechanisms acting at the intraspecific level." would be way more straightforward.

Thank you, reworded.

L298: I would avoid "Unfortunately, however,...", use either one or the other

We kept only "unfortunately" here.

L309: "studies work to distinguish" ...odd wording, rephrase

Rephrased.

L370-383: this paragraph is overlong considering that you state this effect is 'weak' and not confirmed in other subsets of the analyses.

Thank you very much. However, we would like to keep much of our discussion and not exclude discussion of some potentially interesting patterns which we found in our data. We fully acknowledge that we cannot identify the primary drivers of all these patterns. Here we suggest some potential explanations that may inform future analyses.

L384: of course, there is always some association, so strictly speaking this sentence is wrong (no association). You may say that you found very weak, negligible, or non-significant associations, but you should never say that you found no association.

Thank you, we replaced "no" by "negligible".

L397-410: I think this last paragraph does not properly convey the key messages of the study. It is mostly focused on 'applied' issues, whereas this is by far an evolutionary ecology study, investigating broad scale patterns of adaptation of wild species to human disturbance and tolerance of disturbed habitats. That this study was developed "in order to help developing global, evidence-based management interventions" seems out of scope to me, besides being unclear which kind of evidence-based management interventions can be hypothesized based on these results. Perhaps you may explain a bit more (i.e. you may suggest that in an increasingly human dominated world conservation efforts should be more focused on less tolerant species?). In any case, I think you should focus more on the factors promoting adaptation (or lack of adaptation) and its ecological/evolutionary consequences.

This is an interesting point. In the 13 previous paragraphs we discuss the results in detail and some evolutionary ecological implications of the results. We wish, in our final paragraph, to broaden the scope a bit. That said, the comment about which kind of evidence-based management interventions we are implying is a good one and we have added clarity to the paragraph.

In general, I think the discussion is excessively speculative and not focused on the main findings. I think it can be considerably reduced by stating from the beginning that it will focus on the more robust pattern (i.e. those confirmed across multiple datasets).

As described above, we would like to keep much of our discussion, even if somewhat speculative, to stimulate future work.